# Polymer Concentration and Liquid—Liquid Demixing Time Correlation with Porous Structure of Low Dielectric Polyimide in Diffusion-Driven Phase Separation

**DOI:** 10.3390/polym14071425

**Published:** 2022-03-31

**Authors:** Subin Kim, Jaemin Son, Hwon Park, Euigyung Jeong, Ki-Ho Nam, Jin-Seok Bae

**Affiliations:** Department of Textile System Engineering, Kyungpook National University, Daegu 41566, Korea; subin2219@naver.com (S.K.); woals4272@naver.com (J.S.); sd9854@naver.com (H.P.); wolfpack@knu.ac.kr (E.J.)

**Keywords:** polyimide, non-solvent induced phase separation, porous structure, dielectric constant

## Abstract

Porous polyimide (PI) films are a promising low-k dielectric material for high-frequency data transmission with low signal attenuation. Pores are generated by non-solvent induced phase separation (NIPS) during phase inversion of polymer solution via non-solvent accumulation and solvent diffusion. In this study, aromatic PI was employed as a matrix for NIPS, and the influence of polymer concentration and liquid—liquid demixing time on the morphology of pores in the PI films was investigated. This ensured control over the porous structure of the PI film and provided desirable dielectric properties in a broad frequency range of 100 Hz–30 MHz (1.99 at 30 MHz) and thermal stability (*T*_d5%_ > 576 °C, *T*_g_ > 391 °C). This study addresses the effect of polymer concentration and coagulation time on the morphology and physical properties of PI sponge films and provides guidance on the design and optimization of architectures for polymeric materials requiring pore modification.

## 1. Introduction

Nowadays, advances in microelectronics and communication technologies have revolutionized the way we live. With growing demand for extremely high-data-rate services in the wireless communication market, high-frequency and high-speed digital signal transmission, especially upcoming fifth generation (5G) mobile communication technology, has emerged as a hot research topic [1,2]. Aromatic polyimides (PIs) are among the most promising candidates for next-generation interlayer dielectrics [3,4]. In contrast to other commonly applied polymers, such as polybenzobisoxazole, polysilsesquioxane, and SILK (k = 2.65, Dow Chemical), they stand out due to their unique properties. The performance, reliability, and durability of PIs originate from their unique combination of outstanding thermal stability, mechanical strength, low dielectric constant, and good adhesion [5,6]. Unfortunately, owing to the poor polarization of their rigid backbone, the dielectric constant of commercial PIs (k = ~3.5 at 1 kHz) is too high for increasing the data transmission speed of electronic devices [7,8,9]. Typical synthetic routes of porous PIs include: (i) modification of chemical backbone structure (e.g., fluorination, or side-chain substitution) [10,11,12], (ii) incorporation of high-volume air pockets with k = ~1 into the matrix [13,14], and organic–inorganic hybridization [15].

Non-solvent induced phase separation (NIPS) has been commonly applied to prepare a variety of polymeric materials with specific porous morphologies, such as finger- or sponge-like pores [16]. NIPS involves three components (polymer, solvent, and nonsolvent), and the process is induced by interactions between a polymer and a solvent of interest [17]. The rate and duration of diffusive exchange between the solvent and the non-solvent in the polymer plays an important role in the change in composition and in the promotion of certain mechanisms, such as coalescence, resulting in different morphologies. Another important influencing factor is the polymer concentration. It can be significantly influenced by variations in the viscosity of the polymer solution through compositional changes, as the polymer concentration has a high impact on the diffusion rate [18]. Given that the porosity and the type of porous structure of polymer film directly affect its dielectric performance, fine-tuning of phase inversion is very important in the preparation of porous polymer films possessing low-k and low dielectric loss.

The present study focuses on lowering the dielectric constant of PI by porous structure engineering. Herein, we describe a facile approach to prepare porous low-k dielectric PI films via the NIPS process. First, a poly(amic acid) (PAA) intermediate solution was cast on a glass substrate. PAA phase separation proceeded as a diffusive exchange between the solvent and nonsolvent. The hierarchically porous sponge-like PI film was produced after thermal imidization. We optimized PI performance by controlling the independent parameters of polymer concentration and liquid—liquid demixing time. The PI film performance was strongly related to the morphological structure. The enhanced understanding of the phase separation process opens the way to fulfilling user requirements for next-generation interlayer dielectrics.

## 2. Materials and Methods

### 2.1. Materials

Pyromellitic dianhydride (PMDA, >98%) and 4,4′-oxydianiline (ODA, >98%) were purchased from Tokyo Chemical Industry, Co., Ltd. (Tokyo, Japan). 1-Methyl-2-pyrrolidinone (NMP, >99%) and acetone (>99%) were purchased from Duksan Chemical Co., Ltd. (Ansan, Korea) All chemicals were used without further purification.

### 2.2. Preparation of Hierarchically Porous Sponge-like Polyimide Film

The polymerization was carried out by the reaction of stoichiometric equivalent amounts of ODA (0.6 g, 3 mmol) with PMDA (0.65 g, 3 mmol) at a solid concentration of 12–20 wt.% in the NMP. The mixture was stirred for 24 h at 23 °C. A viscous and yellowish poly(amic acid) (PAA) intermediate solution was obtained by the gradual dissolution of PMDA. The PAA solution was coated on a glass substrate using a high-precision adjustable film applicator with a blade gap of 800 μm. The casted PAA films were immersed in a 50 mL of acetone bath for 10–180 min at 23 °C. Subsequently, the opaque PAA films were consecutively cured at 90 °C/2 h, 150 °C/1 h, 200 °C/1 h, 250 °C/30 min, and 300 °C/30 min, resulting in complete conversion of the amic acid group into imide. Finally, the hierarchically porous sponge-like PI films were ripped off the glass substrate by deionized (DI) water and further dried in a convection oven at 50 °C for 8 h. The polymer concentration and liquid—liquid demixing time in the coagulation bath were varied.

### 2.3. Measurements

Scanning electron microscopy (SEM) was performed using a Hitachi SU8220 apparatus at an acceleration voltage of 10 kV. Fourier transform infrared (FTIR) spectra were registered using a Thermo Fisher Scientific (Waltham, MA, USA) Nicolet Is5. XRD was performed on a Panalytical (Malvern, UK) EMPYREAN X-ray diffractometer with Cu Kα radiation (λ = 1.54 Å). Thermogravimetric analysis (TGA) was conducted using a TA Instruments (Delaware, OH, USA) SDT 650 under N_2_ flow at a heating rate of 20 °C min^−1^. Dynamic mechanical analysis (DMA) was performed using a TA Instruments DMA 850 at a heating rate of 3 °C min^−1^ with a load frequency of 1 Hz in air. Dielectric properties were measured with an Agilent (Santa Clara, CA, USA) 4294A Precision Impedance analyzer on the sample mounted between the parallel plate electrodes of a 16451B dielectric test fixture. Compliant with ASTM D150 standard, 16451B has three electrodes, two forming a capacitor and one providing a protective electrode to suppress the effects of ‘fringe capacitance’ and provide accurate measurements. The test fixture was kept in a microprocessor-controlled oven to maintain constant temperature of the sample during the measurement. The dielectric constant (ε′) was calculated using the Formula (1):(1)$$\varepsilon^{\prime }=\frac{C_{m}}{\varepsilon_{0}}\frac{t}{A}=\frac{C_{m}\times t}{\varepsilon_{0}\times \pi \times {\left(d/2\right)}^{2}}$$
where *ε*_0_ is the vacuum permittivity (8.85 × 10^−12^ F m^−1^), *C_m_* is the measured capacitance, *A* is the electrode area, *d* is the electrode diameter, and *t* is the film thickness.

## 3. Results and Discussion

Figure 1 illustrates the overall scheme and a possible mechanism of the preparation of hierarchically porous sponge-like PI films. The procedure included three stages: The homogeneous PAA/NMP complex solution was bar-coated onto a glass substrate. Then the PAA film was precipitated during the solvent-nonsolvent exchange under the chemical potential gradient. The polymer-rich phase formed the film matrix, whereas the dispersive non-solvent-rich phase formed the 3D interconnected micropores. Finally, the solidified porous PAA films were thermally cured at 300 °C, resulting in complete conversion of the amic acid group to imide.

The chemical transformation of PAA to PI was monitored by FTIR (Figure 2a). All films demonstrated characteristic absorption peaks at 1780 cm^−1^ (imide I: asymmetric C=O stretching) and 1720 cm^−1^ (symmetric C=O stretching) [19]. The absorption peak at 1550 cm^−1^ (amide I) disappeared after thermal curing; on the other hand, absorption peaks appeared at 1390 cm^−1^ (imide III: C–N stretching) and at 730 cm^−1^ (imide IV: bending vibrations of cyclic C=O), indicating complete conversion to PI during the thermal dehydration [20].

The average interchain spacing (*d*-spacing) was calculated from the most prominent XRD peak in the glassy amorphous PI spectra. Figure 2b demonstrates that all the PI films consisted of the amorphous phase. However, compared with the nonporous PI film (2θ = 18.6°), the peak of the PI sponge film was slightly shifted to a smaller angle (2θ = 17.6°), that can be attributed to the increased interchain spacing (*d*-spacing) at the site of micropores.

The morphology of the PI films obtained by the NIPS reflects the thermodynamics and phase separation kinetics of the polymer diluent solution. By varying the polymer concentration in the casting solution and liquid—liquid demixing time, the porous structure in fabricated PI films can be easily controlled. The structures of PI sponge films were examined by XRD patterns and SEM micrographs, as shown in Figure 3 and Figure 4. Figure 3a represents the fracture surface of PI films prepared using NMP as a solvent with different PAA concentrations varying from 12 to 20 wt.%. It could be observed that the increase in PAA concentration suppressed the formation of numerous macrovoids (Figure 3b), thus promoting the sponge-like porous structure. At a PAA concentration over 16 wt.%, the sponge-like pore structure was dominant in contrast to the structure obtained at a relatively low concentration. The enhanced viscosity at high polymer concentrations impedes the growth of nuclei responsible for the development of macropores due to the suppression of diffusive exchange and mass transfer between the solvent and nonsolvent during the phase inversion [21,22].

The liquid—liquid demixing time between solvent and nonsolvent is another factor which contributes to the structural transition. Figure 3c shows the fracture surface in the PI films prepared with the same PAA concentration of 20 wt.% as a function of coagulation time. It took approximately 120 min to develop a sponge-like porous structure in the PI films without large cavities. However, large cracks were observed in the fracture surface at coagulation times below 30 min because the phase heterogeneity was insufficient to form a porous structure. The PI sponge films prepared by precipitation for 120 min with 20 wt.% PAA concentration involve a sealed closed-cell surface layer (Figure 4). This observation was consistent with the hypothesis that rapid phase inversion occurred on the outer surface when the PAA solution was directly immersed in a nonsolvent [23]. Therefore, PAA aggregation affecting the formation of a dense surface was detected. A dense layer without pores was formed on the bottom surface. In contrast, interconnected open-cell pores were detected on the bottom surface after 180 min immersion due to excessive solvent leaching from the PAA film.

The dielectric properties were evaluated using a broadband dielectric spectrometer in the frequency range of 100 Hz−30 MHz at room temperature (Figure 5 and Table 1). The dielectric constants (*ε*′) were weakly dependent on frequency over a wide frequency range. The dielectric constant of nonporous PI film was 3.29 and 3.23 at 1 and 30 MHz, respectively, whereas PI sponge film had remarkably lower dielectric constants in all the frequency ranges. Notably, the PI sponge film exhibited an ultralow dielectric constant of 2.0 and 1.99 at 1 and 30 MHz, respectively. This is because the presence of hierarchically porous structure in PI sponge film can introduce air voids, which have an extremely low k value of ~1.0 [24]. Furthermore, dielectric loss in PI sponge film obviously decreased in the high-frequency range. Therefore, the dielectric performance of the resultant PI sponge film may be stable in a broad frequency range from 100 Hz to 30 MHz.

The viscoelastic behaviors of nonporous PI and PI sponge films were investigated with DMA. Figure 6a and Table 1 demonstrate the temperature dependence of the dynamic storage modulus (*E*′) and loss factor (tan δ) for nonporous PI and PI sponge films. The DMA profiles for all the films exhibit glass-to-rubber transitions. The glass transition temperature (*T*_g_) of the films was determined by the tan δ peak position. Nonporous PI film exhibited a single *T*_g_ at 380 °C, whereas the PI sponge film had two relaxation temperatures of 283 °C and 391 °C. The relaxation at 283 °C is a typical microphase separation of porous structures [25] and the distinct peak at 391 °C is associated with segmental mobility. All films have constant *E*′ values before their *T*_g_ appear, reflecting the solid-like character of rigid PI. TGA analysis showed the thermal stability of nonporous PI and PI sponge films, as indicated in Figure 6b and Table 1. Similar thermal degradation behavior was detected in nonporous and PI sponge films. The major thermal degradation with a 5% weight loss (*T*_d5%_) occurred within 564–576 °C. In addition, the amount of the carbonized char residue at 800 °C was >53%. These results demonstrate that the PI sponge film exhibits glass-transition and thermal degradation behaviors similar to that of nonporous PI film and that a complete imidization reaction occurs in the microporous structure [26].

## 4. Conclusions

Hierarchically porous sponge-like PI films were successfully fabricated in this study via NIPS followed by thermal imidization. The influence of polymer concentration and liquid—liquid demixing time on the porous structure of the PI films was investigated. The PAA intermediate solution immersed in an acetone bath initiated NMP diffusion from the PAA solution to a coagulation bath. This phenomenon strongly influenced the mass exchange and phase separation. A hierarchical porous structure was effectively tailored from macrovoids to uniform sponge-like pores by tuning the polymer concentration and coagulation time. These major parameters correlated with the dielectric properties. An ultralow dielectric constant of 1.99 at 30 MHz was achieved in the PI sponge film. The latter also showed appropriate thermal stability (*T*_d5%_ > 576 °C, *T*_g_ > 391 °C). Consequently, optimization of NIPS parameters for phase separation in different types of polymeric materials may provide us with porous materials for many applications.

## Figures and Tables

**Figure 1 polymers-14-01425-f001:**
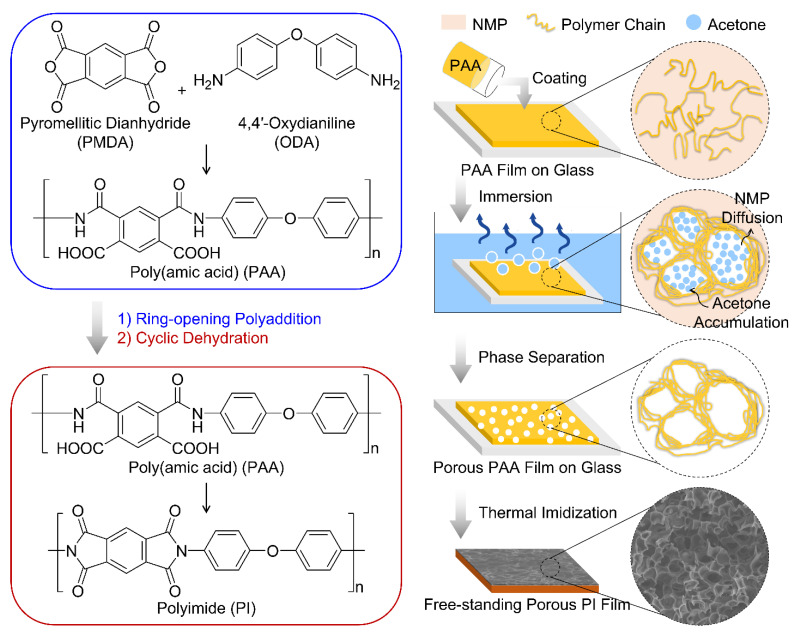
Schematic illustration of the fabrication process of hierarchically porous sponge-like PI films.

**Figure 2 polymers-14-01425-f002:**
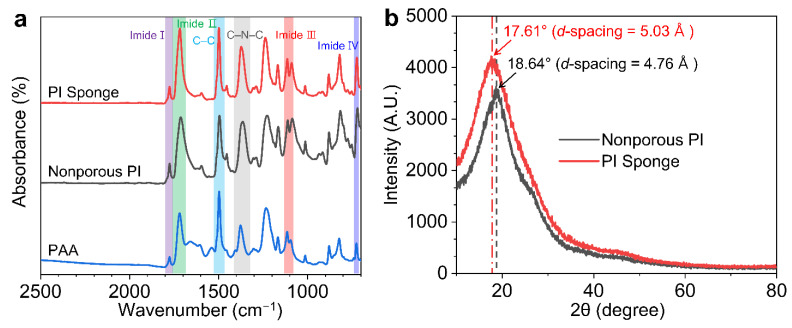
(**a**) FTIR absorption spectra of the PAA, nonporous PI, and PI sponge films. (**b**) XRD patterns of nonporous PI and PI sponge films.

**Figure 3 polymers-14-01425-f003:**
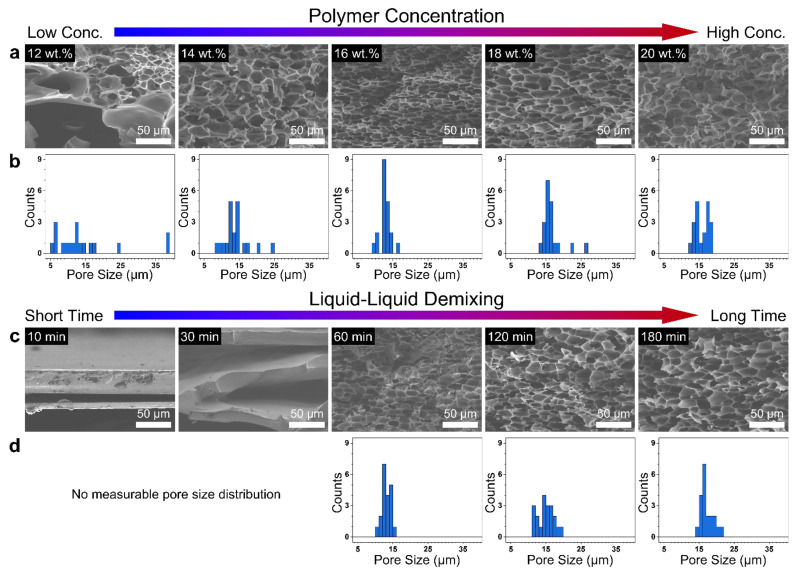
SEM fracture surface morphologies and pore size distributions of the PI films under different (**a**,**b**) polymer concentrations and (**c**,**d**) coagulation times.

**Figure 4 polymers-14-01425-f004:**
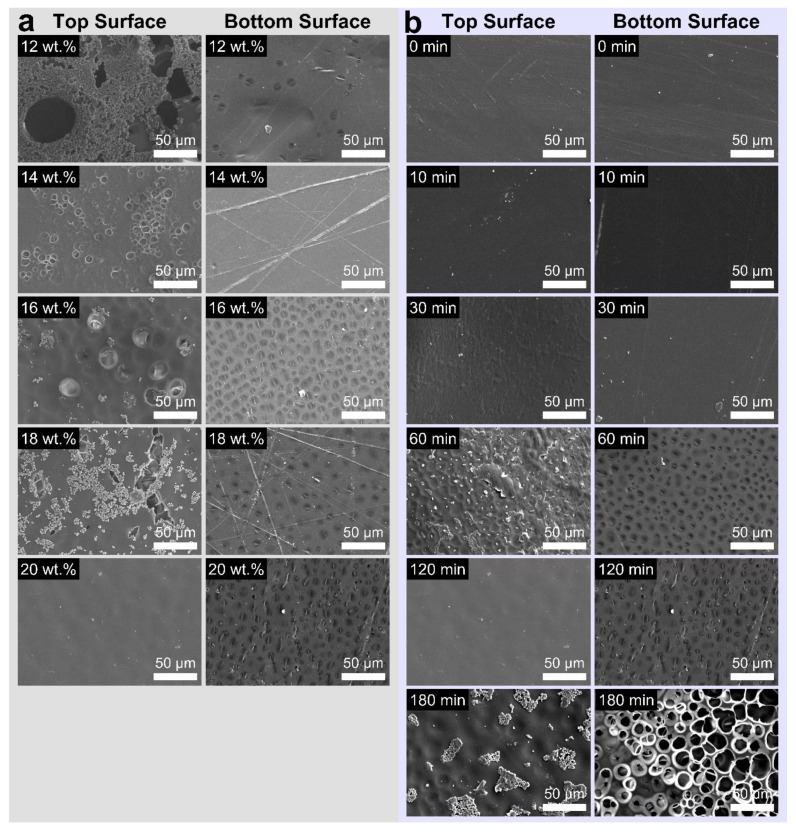
SEM morphologies of the PI films under different (**a**) polymer concentration and (**b**) coagulation time. (**left**) Top surface and (**right**) bottom surface.

**Figure 5 polymers-14-01425-f005:**
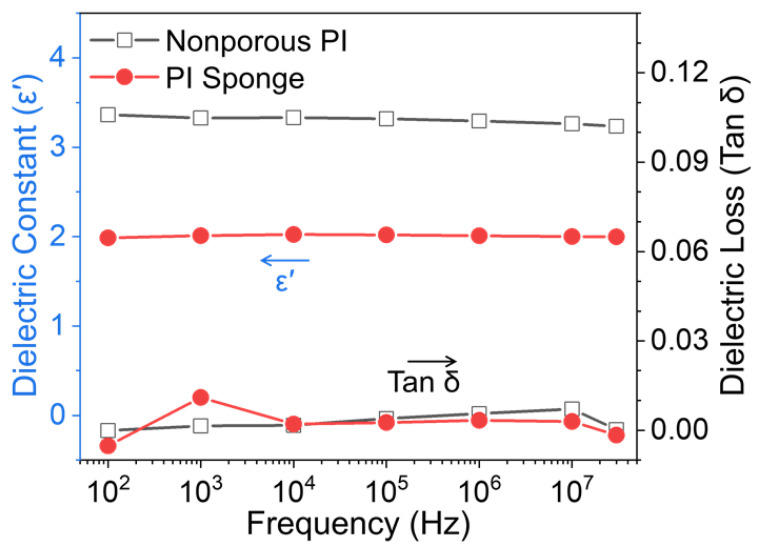
Dielectric constants and dielectric loss of nonporous PI and PI sponge films in the frequency range from 100 Hz to 30 MHz.

**Figure 6 polymers-14-01425-f006:**
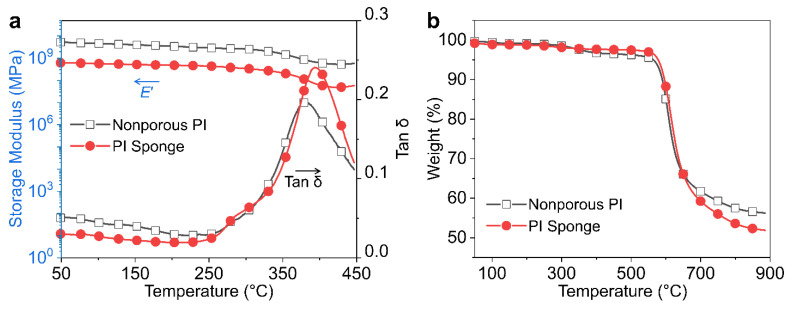
(**a**) DMA and (**b**) TGA curves for nonporous PI and PI sponge films.

**Table 1 polymers-14-01425-t001:** Thermal and dielectric properties of nonporous PI and PI sponge films.

Sample	*T*_g_ (°C)	*T*_d5%_ (°C)	*T*_d10%_ (°C)	Char Residue at 800 °C (%)	Dielectric Constant at 1 MHz	Dielectric Loss at 1 MHz
Nonporous PI film	381	564	589	57.47	3.29	5.60
PI Sponge film	392	577	596	53.56	2.01	3.34

## Data Availability

Not applicable.

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
