# Peer review of "Polymer Concentration and Liquid—Liquid Demixing Time Correlation with Porous Structure of Low Dielectric Polyimide in Diffusion-Driven Phase Separation"

_polymers, 2022, doi:10.3390/polym14071425_

Round 1

Reviewer 1 Report

This is an interesting paper in which porous low-k dielectric PI films were prepared. The authors investigated the effect of polymer concentration and liquid–liquid demixing time on the performances of the aromatic polyimide film. I examined the revised version of the manuscript and I accept the publication in the present form.

Author Response

We appreciate the positive decision of the reviewer for publication of our manuscript in Journal Polymers.

Reviewer 2 Report

Dear Authors,

The manuscript is ready to accept. The authors have made changes to the suggestion.

Author Response

(The authors gave the same response as above.)

Reviewer 3 Report

This paper has been revised and greatly improved. However, there are still some problems that need to be modified.

  1. The letters in Eq. (1) in Section 3 should be consistent with the letter symbols in the explanation below.
  2. In Section 3, there some errors in the expression of Figure 5, which shows the change of the dielectric constant of PI film between 100Hz and 30MHz. The dielectric constant of PI film without hole is 3.29 at 100Hz, instead of 1Hz. It is suggested to check and correct it carefully.

Author Response

Author reply 1: We thank the reviewer for valuable comment. In the Result Section, we clearly indicate the symbol of the dielectric constant as shown below to prevent the reader from confusing the storage modulus (E’) and dielectric constant (ε’).

Added/Revised text [Manuscript, page 3]: The dielectric constant (ε’) was calculated using the formula (1):

Added/Revised text [Manuscript, page 6]: The dielectric constants (ε’) were weakly dependent on frequency over a wide frequency range.

Author reply 2: As the reviewer commented, the dielectric properties of the PI films were measured in the range of 100 Hz to 30 MHz. The 3.29 of the nonporous PI film mentioned in the Result Section is a dielectric constant at 1 MHz, not 1 Hz.

This manuscript is a resubmission of an earlier submission. The following is a list of the peer review reports and author responses from that submission.

Round 1

Reviewer 1 Report

It is interesting to study the polymer concentration and liquid-liquid demixing time correlation with porous structure of low dielectric polyimide in diffusion-driven phase separation. However, there are many problems that need to be revised. 1 The paper simply introduces some simple experimental phenomena without explaining the reasons. The authors need to give a detailed analysis. 2 There are some problems in the paper structure. For example, the paper has three 2.1 parts. 3 The 67th line is "see Supplementary figure S1", and the paper does not contain figure S1. 4 In line 103, "as given in Figure S2 and Figure 2", the paper does not contain the figure number of figure S2. 5 In Figure 2, A, B, C and D are marked in the figure, but the title of Figure 2 and many parts of the paper are marked as a, b, c and d in Figure 2.

Reviewer 2 Report

Hello Author,

The author describes the synthesis of polyimide and casting method by using solven and co-solvent to make porous polyimide. Here are my comments

(1) What is the novelty of this work. Why particular use of polyimide is used to get low dielectric material. There is another material in literature. The author failed to explain how their material and method are unique.

(2)The synthesis method to close the rings involve curing at a much higher temperature and inert atmosphere. Why not use acetic anhydride to get polyimide has already been well established in the literature. 

(3) What is the glass transition temperature of porous material. Why there is a shift in tan delta value for porous and non-porous material

Reviewer 3 Report

This is an interesting paper (communication-type) in which porous low-k dielectric PI films were prepared. The authors investigated the effect of polymer concentration and liquid–liquid demixing time on the performances of the aromatic polyimide film. Overall, the manuscript is written in clear English. I recommend minor revision. Please see the questions and comment below.

1) The electrical contact is very important for evaluation of dielectric properties of a material. How the authors prepared the electrodes on both sides of the samples.

2) Nothing was mentioned about how many replicates were used in each experiment and the standard deviation of the results. The values of dielectric constant are too accurate. In my opinion, the values should be reconsidered to maximum one decimal place accuracy.